# Fe_3_O_4_@TiO_2_ Microspheres: Harnessing O_2_ Release and ROS Generation for Combination CDT/PDT/PTT/Chemotherapy in Tumours

**DOI:** 10.3390/nano14060498

**Published:** 2024-03-10

**Authors:** Bo Zhao, Xiuli Hu, Lu Chen, Xin Wu, Donghui Wang, Hongshui Wang, Chunyong Liang

**Affiliations:** 1School of Materials Science and Engineering, Hebei University of Technology, Tianjin 300401, China; z15079039978@163.com (B.Z.); a1327052783@163.com (L.C.); 13131952361@163.com (X.W.); 2Institute of Polymer Science and Engineering, School of Chemical Engineering, Hebei University of Technology, Tianjin 300130, China; huxiuli@hebut.edu.cn; 3Center for Health Science and Engineering, Hebei Key Laboratory of Biomaterials and Smart Theranostics, Tianjin 300131, China; donghuiwang@hebut.edu.cn

**Keywords:** PTT, PDT, CDT, cancer therapy, magnetic field

## Abstract

In the treatment of various cancers, photodynamic therapy (PDT) has been extensively studied as an effective therapeutic modality. As a potential alternative to conventional chemotherapy, PDT has been limited due to the low Reactive Oxygen Species (ROS) yield of photosensitisers. Herein, a nanoplatform containing mesoporous Fe_3_O_4_@TiO_2_ microspheres was developed for near-infrared (NIR)-light-enhanced chemodynamical therapy (CDT) and PDT. Titanium dioxide (TiO_2_) has been shown to be a very effective PDT agent; however, the hypoxic tumour microenvironment partly affects its in vivo PDT efficacy. A peroxidase-like enzyme, Fe_3_O_4_, catalyses the decomposition of H_2_O_2_ in the cytoplasm to produce O_2_, helping overcome tumour hypoxia and increase ROS production in response to PDT. Moreover, Fe^2+^ in Fe_3_O_4_ could catalyse H_2_O_2_ decomposition to produce cytotoxic hydroxyl radicals within tumour cells, which would result in tumour CDT. The photonic hyperthermia of Fe_3_O_4_@TiO_2_ could not only directly damage the tumour but also improve the efficiency of CDT from Fe_3_O_4_. Cancer-killing effectiveness has been maximised by successfully loading the chemotherapeutic drug DOX, which can be released efficiently using NIR excitation and slight acidification. Moreover, the nanoplatform has high saturation magnetisation (20 emu/g), making it suitable for magnetic targeting. The in vitro results show that the Fe_3_O_4_@TiO_2_/DOX nanoplatforms exhibited good biocompatibility as well as synergetic effects against tumours in combination with CDT/PDT/PTT/chemotherapy.

## 1. Introduction

In our modern society, cancer is one of the most serious diseases [1]. In cancer disease, ROS play an important role, as light offers distinct advantages, including minimal invasiveness and spatiotemporal resolution [2,3]. Through irradiation with lasers, PDT can result in the death of cancer cells by generating cytotoxic ROS from photosensitisers (PS) [4]. During PDT, PS drugs generate ROS through a series of photochemical reactions to induce cytotoxicity [5]. Traditional organic photosensitisers, such as small-molecule organic PSs like sulphonamide–cyanine dye 7 (Cy7), indocyanine green (ICG), and porphyrins, can be utilised as PSs owing to their remarkable efficacy in generating high levels of ROS. Nonetheless, their potential biomedical applications encounter limitations due to factors such as inadequate photostability and sensitivity constraints influenced by environmental factors [6]. In contrast, inorganic materials such as TiO_2_ have shown long-term physical and chemical stability [7]. TiO_2_ has a large band gap energy (3.0 eV for anatase phase and 3.2 eV for rutile phase). PDT relies on the formation of an electron–hole pair upon the absorption of a photon with an energy equal to or higher than the semiconductor’s band gap [8]. When combined with oxygen or water, these lone electrons form hydroxyl radicals (•OH), hydrogen peroxides (H_2_O_2_), and super oxides (O_2_^−^). The generated ROS can attack virtually all macromolecules, which results in serious damage to cancer cellular components [9]. However, the low ROS generation efficacy and the lack of targeting mechanisms remain critical issues in achieving a desirable tumour-suppressing effect and favourable therapeutic outcome [10,11]. Moreover, low oxygen levels in hypoxic tumour microenvironments (TMEs) result in low ROS production and might result in poor therapy outcomes for PDT [12,13,14,15,16]. Furthermore, tumour cells are rich in glutathione (GSH), which enables the ^1^O_2_ produced during PDT to be rapidly reduced, thus diminishing the efficacy of PDT [17,18,19,20,21,22]. These two TME-associated factors greatly limit the efficacy of PDT [23].

The magnetic properties, adjustable size, biocompatibility, and easier synthesis of Fe_3_O_4_ nanoparticles make them highly attractive for cancer therapies compared to other magnetic materials [24,25,26]. Fe_3_O_4_ nanoparticles demonstrate inherent pH-responsive peroxidase (POD)- and catalase (CAT)-like properties, enabling them to facilitate the breakdown of H_2_O_2_ into oxygen and •OH, respectively [24,25,26]. When Fe_3_O_4_ reacts with H_2_O_2_ at neutral pH, it can produce water and oxygen, but in an acidic TME, it can produce toxic ROS and •OH [27,28,29,30].

In order to improve the efficiency of PDT, we successfully prepared a novel composite photocatalytic material (Fe_3_O_4_@TiO_2_/DOX) composed of the typical photosensitiser TiO_2_ and enzyme Fe_3_O_4_ components (Figure 1). The Fe_3_O_4_ component’s catalase-like function can facilitate the breakdown of surplus H_2_O_2_, generating O_2_ within the tumour’s cytoplasm to improve PDT [12,31,32,33,34]. Fe_3_O_4_ can slowly release Fe^2+^ and Fe^3+^, induced by the low-pH environment of tumour cells [35]. H_2_O_2_ reacts with Fe^2+^ to form Fe^3+^ and a hydroxyl radical to kill tumour cells. In detail, Fe^3+^ doping can transform GSH to glutathione disulphide (GSSH), which decreases the consumption of ROS [23]. Remarkably, the heat generated by the photothermal conversion of Fe_3_O_4_@TiO_2_ further promotes the efficiency of Fenton-like and photocatalysis reactions. The incorporation of DOX into Fe_3_O_4_@TiO_2_ microspheres enhances targeted delivery to cancer cells. Concurrently, these microspheres facilitate the controlled and sustained release of DOX, optimising therapeutic efficacy while mitigating potential side effects.

## 2. Materials and Methods

### 2.1. Preparation of Fe_3_O_4_ Microspheres

The synthesis of the Fe-glycerate precursor involved mixing Fe(NO_3_)_3_∙6H_2_O (AR, Macklin., Shanghai, China), glycerol (AR, KERMEL, Tianjin, China), and isopropanol (AR, KERMEL, Tianjin, China) under high-temperature and high-pressure conditions. This process included ultrasonic treatment, dropwise addition into deionised water with vigorous stirring, a 12 h reaction in a stainless-steel high-pressure vessel lined with polytetrafluoroethylene at 190 °C, followed by centrifugation, multiple washes with deionised water and ethanol, and drying for 24 h at 60 °C, while purging air with argon. Subsequently, the Fe-glycerol precursor was heated at a ramp rate of 2 °C min^−1^ under an argon atmosphere to 500 °C for 3 h, yielding the desired Fe-glycerate substance. These procedures encompassed chemical mixing, high-temperature high-pressure reactions, and subsequent processing steps, ensuring the purity and desired form of the final product.

### 2.2. Preparation of Fe_3_O_4_@TiO_2_ Microspheres

In this experimental procedure, 0.3 g of Pluronic F127 (with a molecular weight of 12,600) (AR, Macklin., Shanghai, China) were introduced into 50 mL of water while being stirred continuously for a duration of 200 min. Following this, 60 mg of previously prepared Fe_3_O_4_ microspheres were added to the mixture and mechanically stirred for an additional 30 min. Subsequently, 1.5 mL of tetra butyl titanate (AR, Macklin., Shanghai, China) were slowly incorporated into the mixture and stirred for a duration of 1 h. Following the completion of the stirring process, the Fe_3_O_4_@TiO_2_ core-shell structures were obtained. To conclude the procedure, the obtained Fe_3_O_4_@TiO_2_ core-shell structures were separated via centrifugation and washed multiple times with ethanol to eliminate impurities or residual compounds.

### 2.3. Structural Characterization

X-ray diffraction (XRD) patterns were obtained using a Siemens D 5005 X-ray (Siemens, Munich, Germany) diffractometer equipped with Cu Kα radiation (40 kV, 30 mA). Particle morphology was analysed via both scanning electron microscopy (SEM) and transmission electron microscopy (JEOL JEM-2100, Tokyo, Japan). Nitrogen adsorption–desorption isotherms were measured using a Micromeritics Tristar 3000 (Micromeritics Instrument Corporation, Norcross, GA, USA) to determine the Brunauer–Emmet–Teller (BET) surface area and porosity based on the Barrett–Joyner–Halenda (BJH) model, including the single-point total pore volume calculated at P/P0 = 0.98. The successful immobilization of TiO_2_ onto the Fe_3_O_4_ microspheres was confirmed using X-ray photoelectron spectroscopy (XPS). Zeta potential analysis (SURPASS 3, Anton Paar GmbH, Graz, Austria) determined the surface charge. Fourier transform infrared spectroscopy (FT-IR, TENSOR 27, Bruker, Billerica, MA, USA) was employed to identify functional groups. UV–Vis spectroscopy (UV-6100, Mapada, Shanghai, China) was conducted to analyse the spectra.

### 2.4. DOX Loading and Release

The encapsulation process of DOX (Doxorubicin) (AR, Macklin., Shanghai, China) into Fe_3_O_4_@TiO_2_ involved dissolving 10 mg of DOX in a 10 mL PBS (pH 7.4) solution and suspending 50 mg of Fe_3_O_4_@TiO_2_ in this solution. The mixture was stirred in darkness for 24 h, followed by the collection of the supernatant for UV–vis spectrophotometry analysis at 490 nm. The drug loading content within the Fe_3_O_4_@TiO_2_ nanoplatform was determined. This encapsulation method assesses the efficiency of DOX binding to the nanoplatform, essential for evaluating its potential in targeted drug delivery and therapeutic applications. DOX loading efficiency was calculated as below:Drug loading efficiency (LE) = (Weight of loaded DOX (mg)/Weight of total DOX (mg)) × 100(1)
Drug loading capacity = (total amount of DOX − Free DOX)/amount of Fe_3_O_4_@TiO_2_/DOX(2)

The in vitro investigation focused on the pH responsiveness and drug release characteristics of DOX-loaded formulations in PBS buffers with varying pH levels (pH 5.4 or 7.4). The study involved dispersing 5 mg of DOX-loaded microparticles (Fe_3_O_4_@TiO_2_) separately into 10 mL volumes of acetate buffers with differing pH values (5.4 or 7.4). The quantification of released DOX in the supernatant was conducted using a UV–visible (UV–vis) spectrophotometer. Following irradiation with a NIR laser for 10 min, the UV–vis absorption spectra of the supernatant were recorded to assess photothermal-triggered drug release.
$$DOX\;release(\%)=\sum_{i=1}^{n}M_{i}/M_{0}\times 100\%$$
where $M_{i}$ is the amount of DOX released from Fe_3_O_4_@TiO_2_/DOX at time and $M_{0}$ is the total amount of loaded drug in Fe_3_O_4_@TiO_2_.

### 2.5. NIR-Triggered Photothermal Effect of Fe_3_O_4_@TiO_2_ Microspheres

To assess the photothermal properties, varying concentrations of Fe_3_O_4_@TiO_2_ microspheres were introduced into a cuvette and subjected to irradiation using an 808 nm laser at different power densities. The study also investigated the photothermal effect of Fe_3_O_4_@TiO_2_ under distinct laser powers (1, 1.5, and 2 W/cm^2^).

### 2.6. Intracellular ROS Detection

HeLa (pricella, Wuhan, China) cells were seeded in a 96-well plate at a density of 5 × 10^3^ cells/100 μL per well and incubated overnight. Then, the medium was removed and replaced with a medium containing the Fe_3_O_4_@TiO_2_. After treatment, cells were incubated with 10 μM of DCFH-DA at 37 °C for 20 min and then washed twice with PBS. Cells were irradiated for 10 min with UV light of 365 nm. Afterwards, the cell nucleus was stained with DAPI. Finally, the fluorescence images were observed via CLSM (λ ex = 488 nm, λ em = 525 nm).

### 2.7. Cellular Cytotoxicity Assay

The cytotoxicity of the Fe_3_O_4_@TiO_2_ and Fe_3_O_4_@TiO_2_/DOX was analysed using the CCK-8 assay. After incubating cells in a culture medium within 96-well plates for 24 h, they were subsequently categorised into distinct groups and exposed to varying samples for treatment. For CCK-8 detection, 10 μL CCK-8 reagent was added to the culture medium 4 h before analysis.

### 2.8. Evaluation of •OH Generation

According to MB degradation under the oxidative conditions, a classic colorimetric method was adopted to detect the production of •OH. Briefly, Fe_3_O_4_@TiO_2_ was added into an H_2_O_2_ (10 mmol/L) aqueous solution containing MB (15 μg/mL) and GSH (5 mmol/L). The laboratory shaker was employed for incubating the mixture at a speed of 300 rotations per minute (rpm) and a temperature of 37 °C for a duration of 4 h. Subsequently, the determination of the concentration of the MB solution was performed by measuring the absorbance at 644 nm.

### 2.9. Singlet Oxygen Detection

In this experiment, a solution of 1,3-Diphenylisobenzofuran (DPBF) is prepared by dissolving it in dimethylformamide (DMF) with a concentration of 0.6 mg mL^−1^. The DPBF solution is then mixed separately with Fe_3_O_4_@TiO_2_, maintaining a Fe_3_O_4_@TiO_2_ concentration of 0.5 mg mL^−1^. All mixing steps are carried out meticulously under light-free conditions to prevent interference from external light sources. The UV absorbance of the solution is systematically recorded at 420 nm at various time points over a 40 min duration, utilizing UV–vis spectra analysis for accurate measurement. The experimental design is implemented rigorously to ensure controlled conditions, and the reaction kinetics are monitored accurately to comprehensively investigate the photochemical interactions between DPBF and Fe_3_O_4_@TiO_2_.

### 2.10. Cell Culture and Intracellular Localization Assay

HeLa Cells (5 × 10^3^ cells/well) were seeded onto 96-well plate and incubated overnight under 5% CO_2_ at 37 °C. The fresh medium containing Fe_3_O_4_@TiO_2_-FITC microspheres and Fe_3_O_4_@TiO_2_-FITC with a magnet (the magnet is positioned beneath the 96-well plate) were added and cultivated for another 6 h. Subsequently, observation was conducted using a fluorescence microscope.

In this experimental protocol, Fe_3_O_4_@TiO_2_-FITC microspheres were synthesised by dissolving 20 mg of Fe_3_O_4_@TiO_2_ in 20 mL of PBS and incorporating 1 mg of FITC, followed by overnight stirring in darkness.

## 3. Results and Discussions

The procedure for synthesising Fe_3_O_4_@TiO_2_/DOX microspheres is illustrated in Figure 1a. In the primary stage, TiO_2_ coatings are applied onto Fe_3_O_4_ microspheres using tetrabutyl titanate. Subsequently, the resulting Fe_3_O_4_@TiO_2_ microspheres undergo a 24 h stirring process in a DOX solution, facilitating the effective loading of DOX into the Fe_3_O_4_@TiO_2_ microspheres. The preparation of Fe_3_O_4_ microspheres was carried out in two steps. Hollow spheres containing Fe were first synthesised using the hydrothermal method at 190 °C with Fe(NO_3_)_3_·6H_2_O as a precursor. Upon annealing at 500 °C, the Fe-containing spheres were completely converted to crystalline Fe_3_O_4_ without other phases. These Fe_3_O_4_ microspheres are hollow structures assembled via interlaced nanosheets (Figure 1b,c). The diameters of the Fe_3_O_4_ hollow spheres were calculated to be 600–1200 nm using Nano Measurer measurements (Figure 1d). The SEM image of Fe_3_O_4_@TiO_2_ reveals that the surface of the Fe_3_O_4_ is coated, with the nanosheets of Fe_3_O_4_ being enveloped (Figure 1e). The SEM and TEM images of Fe_3_O_4_@TiO_2_ confirmed the uniform spherical morphology and hollow interior with a diameter of approximately 600–12,000 nm, as shown in Figure 1e–g. These diameters were almost in agreement with those measured from the SEM and TEM images.

The phase characteristics of the Fe_3_O_4_ and Fe_3_O_4_@TiO_2_ architecture were surveyed using XRD (Figure 2a). According to the XRD pattern of Fe_3_O_4_@TiO_2_, there is a combination of the (220), (311), (400), (511), and (440) peaks of Fe_3_O_4_ and the (101), (004), and (200) peaks of anatase TiO_2_. Based on the SEM and EDS tests, the elemental compositions of the Fe_3_O_4_ microspheres and the Fe_3_O_4_@TiO_2_ composite microspheres were determined. The EDS mapping in Figure 2b illustrates the distribution of O, Fe, N, and C within the Fe_3_O_4_ microspheres, providing a visual representation of their elemental presence. Moreover, the EDS of the Fe_3_O_4_@TiO_2_ composite microspheres (Figure 2c) distinctly reveals the predominance of C, N, Ti, and Fe. The elemental analysis concludes that the Fe_3_O_4_ composite microspheres contain O, Fe, N, and C. The presence of C may be attributed to the addition of glycerol and isopropanol during the hydrothermal reaction process, as well as its persistence in the form of C after annealing at 500 °C. The trace amount of nitrogen, N, is a result of the introduction of Fe(NO_3_)_3_·6H_2_O. The contents of each element are illustrated in Table 1. Additionally, the Fe_3_O_4_@TiO_2_ composite microspheres primarily comprise C, N, Ti, and Fe, indicating the chemical composition of the sample. Notably, titanium (Ti) was identified as a significant component within the composite microspheres, suggesting its potential existence as a TiO_2_ coating. The zeta potential of Fe_3_O_4_ and Fe_3_O_4_@TiO_2_ was also tested, and a higher negative zeta potential of Fe_3_O_4_ after TiO_2_ coating was observed (3.72 mV versus 5.16 mV) (Figure 2d). The change in potential may be due to titanium dioxide coating the surface of ferric oxide. Based on the XPS spectra, the Ti 2p XPS spectrum displays prominent peaks at 464.18 eV (Ti 2p_1/2_) and 458.48 eV (Ti 2p_3/2_), indicative of the presence of Ti in the Ti^4+^ oxidation state within the Fe_3_O_4_@TiO_2_ microspheres (Figure 2i). Furthermore, the analysis of the Fe_3_O_4_@TiO_2_ microspheres revealed distinct peaks at 709.57 and 723.20 eV for Fe 2p_3/2_ and Fe 2p_1/2_ of Fe^2+^, in addition to peaks at 715.4 eV and 728.1 eV corresponding to Fe 2p_3/2_ and Fe 2p_1/2_ of Fe^3+^ (Figure 2h). These results confirm the coexistence of Fe_3_O_4_ and TiO_2_ within the Fe_3_O_4_@TiO_2_ microspheres.

The evaluation of ROS generation from Fe_3_O_4_@TiO_2_ was performed using various chemical probes, as depicted in Figure 3a. Under UV irradiation, the production of ^1^O_2_ by Fe_3_O_4_@TiO_2_ was assessed using DPBF (1,3-diphenylisobenzofuran). While the free DPBF exhibited no significant alteration in its UV–Vis absorbance spectra, Fe_3_O_4_@TiO_2_ displayed a rapid decrease in DPBF absorption upon UV laser exposure, indicating a swift generation of ^1^O_2_ (as shown in Figure 3b,c). Furthermore, the observed colour changes of the methylene blue (MB) solution from blue to colourless over time suggested the effective generation of •OH by Fe_3_O_4_@TiO_2_ (Figure 3d). This phenomenon was attributed to the Fenton-like effect arising from the interaction between released iron ions and H_2_O_2_. The study also uncovered pH-dependent enzymatic activities of Fe_3_O_4_. Under acidic conditions, it exhibited peroxidase activity, leading to the generation of highly toxic •OH after reacting with H_2_O_2_. Conversely, under neutral conditions, it displayed catalase-like activity, facilitating the decomposition of H_2_O_2_ into harmless H_2_O and O_2_ [36,37,38]. Moreover, an investigation using dissolved O_2_ apparatus demonstrated that Fe_3_O_4_@TiO_2_ possessed catalase-like properties at pH 7.4, promoting the decomposition of H_2_O_2_ to produce O_2_, as illustrated in Figure 3e. In summary, the comprehensive findings highlight the versatile ROS-generating capabilities of Fe_3_O_4_@TiO_2_, emphasising its potential in the field of tumour therapy. Furthermore, its capacity to generate oxygen by decomposing H_2_O_2_ adds significant value to its potential applications in various aspects of tumour treatment.

We then explored the photothermal performance (PTT) of Fe_3_O_4_@TiO_2_. Figure 4a depicts the absorption spectra of Fe_3_O_4_ and Fe_3_O_4_@TiO_2_ microspheres in the UV–visible range. Both Fe_3_O_4_ and Fe_3_O_4_@TiO_2_ microspheres exhibited excellent PTT, as shown in Figure 4b. The TiO_2_ coating did not obviously weaken the photothermal effect of Fe_3_O_4_. The temperature increased to 60 °C and 65 °C after 5 min of irradiation for the Fe_3_O_4_ and Fe_3_O_4_@TiO_2_ suspension, respectively (Figure 4b). In the PTT experiments, it was observed that as the concentration of Fe_3_O_4_@TiO_2_ microspheres increased, there was a substantial elevation in the temperature of the solution (Figure 4c). Further, the solution temperature rise was found to increase with the power of the laser beam (Figure 4d). The photothermal stability of Fe_3_O_4_@TiO_2_ was further evaluated through several photothermal cycles. The evaluation after five ON/OFF laser cycles indicated the exceptional photothermal stability of Fe_3_O_4_@TiO_2_, showcasing negligible alterations in its performance (Figure 4e). Assessments using infrared thermal imaging displayed temperature increments upon exposure to NIR laser radiation. Figure 4f illustrates the gradual temperature increase in Fe_3_O_4_@TiO_2_ (1 mg mL^−1^, 1 W cm^2^) when exposed to 808 laser irradiations. In comparison, Fe_3_O_4_ demonstrated a temperature change of 35.3 °C, whereas Fe_3_O_4_@TiO_2_ showed a temperature change of 29.2 °C under identical NIR irradiation conditions (Figure 4g). Moreover, the calculated photothermal conversion efficiencies of Fe_3_O_4_ and Fe_3_O_4_@TiO_2_ were 65.56% and 54.56%, respectively (Figure 4h).

The saturation magnetisation (MS) and coercive field (Hc) at room temperature are presented in Figure 4i. Both materials exhibit typical superparamagnetic characteristics. The MS and Hc values for Fe_3_O_4_ are 23.1 emu/g and 58.75 Oe, respectively, while those for Fe_3_O_4_@TiO_2_ are 4.45 emu/g and 55 Oe (Figure 4c). The reduction in magnetic saturation is attributed to the introduction of non-magnetic TiO_2_ during synthesis. Additionally, the homogeneously dispersed Fe_3_O_4_@TiO_2_ rapidly adhered to the vial walls, resulting in a clear and transparent solution upon the application of an external magnetic field (MF). Subsequently, upon removal of the magnets, the Fe_3_O_4_@TiO_2_ microspheres uniformly dispersed again after gentle shaking, demonstrating their excellent water-dispersive capability. This magnetic responsiveness holds promise for potential applications in magnetically targeted tumour therapy.

N_2_ adsorption–desorption isotherm analysis was carried out to measure the specific surface area and pore size distribution to discover the BET-specific surface area and the BJH pore size distribution for the as-synthesised Fe_3_O_4_@TiO_2_ (Figure 5a,b). The Fe_3_O_4_@TiO_2_ microspheres, with an extensive surface area measuring 84.611 m^2^/g and a pore size distribution within the range from 1 to 2 nm, serve as facilitating factors for promoting and enhancing the loading of drugs. The typical chemotherapeutic drug DOX was selected to evaluate Fe_3_O_4_@TiO_2_ as a carrier. Compared to the Fourier infrared absorption spectrum of Fe_3_O_4_@TiO_2_, a red shift was noted in Fe_3_O_4_@TiO_2_/DOX, suggesting the effective incorporation of the chemotherapeutic drug DOX into Fe_3_O_4_@TiO_2_ microspheres (Figure 5c). The concentration of DOX in the supernatant was calculated based on the fitting curve shown in Figure 5d. The LC of DOX in Fe_3_O_4_@TiO_2_ is 32.2%. The LE of DOX in Fe_3_O_4_@TiO_2_ is 85.6%.

Therefore, Fe_3_O_4_@TiO_2_/DOX is shown to be a promising chemo-photothermal nanocarrier due to its capacity for controlled drug release responsive to dual stimuli (pH and NIR). The Fe_3_O_4_@TiO_2_/DOX microspheres demonstrate both pH-responsive and photo-responsive release properties. Lowering the pH from 7.4 to 5.4 notably increases the rate and quantity of DOX release, with a significant difference observed between pH 5.5 and pH 7.4, releasing 42% and 19% of DOX, respectively (Figure 5e). NIR irradiation further accelerates the overall release rate of DOX (Figure 5f). Therefore, Fe_3_O_4_@TiO_2_/DOX is a promising chemo-photothermal nanocarrier due to its capacity for controlled drug release and its response to dual stimuli (pH and NIR).

Figure 6 depicts CLSM images of HeLa cells incubated with Fe_3_O_4_@TiO_2_-FTTC microspheres for 8 h, showcasing distinct blue fluorescence from DAPI staining for cell nuclei and green fluorescence from labelled FITC. The Fe_3_O_4_@TiO_2_ was magnetically manipulated by the external magnetic field, showing an obvious green fluorescence signal overlapping with the blue fluorescence signal. The Fe_3_O_4_@TiO_2_ group without magnetic field loading exhibited a weak FITC fluorescence signal at 8 h co-incubation. The results presented in Figure 6 demonstrate that the application of a magnetic field enhances the cellular uptake of the Fe_3_O_4_@TiO_2_.

The cytotoxicity assessment of Fe_3_O_4_@TiO_2_ was conducted via a CCK-8 assay, involving various concentrations and times of co-culture with HeLa cells. As shown in Figure 7a, even at the highest concentration, the cell viability was more than 90%. This observation signifies the notable biocompatibility of Fe_3_O_4_@TiO_2_, suggesting its potential as a safe and efficient drug delivery platform.

Photothermal therapy experiments were performed on HeLa cells to assess the effectiveness of Fe_3_O_4_@TiO_2_ in inducing cellular heat using light. The cell viability decreased to 48.99% after 5 min with a 1.5 W/cm^2^ NIR laser (Figure 7b). Thus, Fe_3_O_4_@TiO_2_ may have good application prospects for photothermal therapy. Next, we studied the PDT effects of Fe_3_O_4_@TiO_2_ on HeLa cells in vitro. After 30 min of ultraviolet (UV) light exposure, the cell viability notably decreased from 81.06% to 52.24%, demonstrating the remarkable performance of the material as a photosensitiser. Given the H_2_O_2_ concentration in tumour tissues ranging from 50 to 100 µM [39,40,41], a concentration of 100 μM of H_2_O_2_ was chosen for subsequent assays related to CDT. As shown in Figure 7b, HeLa cells with exposure to Fe_3_O_4_@TiO_2_ + H_2_O_2_ exposited a significant decrease in viability compared to that of the Fe_3_O_4_@TiO_2_ group cells. Moreover, a CCK-8 assay was conducted to assess the combined therapeutic efficacy of Fe_3_O_4_@TiO_2_/DOX, capable of combining PTT, PDT, CDT, and chemotherapy to target tumour cells. The in vitro experiments conducted on HeLa cells using Fe_3_O_4_@TiO_2_/DOX demonstrated exceptional efficiency in tumour cell elimination. When HeLa cells treated with Fe_3_O_4_@TiO_2_/DOX (1 mg mL^−1^) were exposed to the combined PTT + PDT + CDT + chemotherapy group (Fe_3_O_4_@TiO_2_/DOX + UV + H_2_O_2_ + 808 nm), their survival rate notably decreased to 7.8%. This rate was lower than that of the PDT group (Fe_3_O_4_@TiO_2_/DOX + UV), CDT group (Fe_3_O_4_@TiO_2_/DOX + H_2_O_2_), and PTT group (Fe_3_O_4_@TiO_2_/DOX + 808 nm). Therefore, the construction of Fe_3_O_4_@TiO_2_/DOX microspheres is capable of achieving the combination treatment of PTT, PDT, CDT, and chemotherapy. As depicted in Figure 7c, the cytotoxic effect of the Fe_3_O_4_@TiO_2_/DOX + 808 + H_2_O_2_ group on HeLa cells closely approximates that of the free DOX group. These findings underscore the potential advantages of multimodal therapeutic strategies in cancer treatment. To verify the effects of the H_2_O_2_, UV, and 808 nm laser used in the experiment on cells, cells were treated with 100 μM H_2_O_2_, UV, and 808 nm lasers, and their survival rates were measured using a CCK-8 assay (Figure 7d).

Here, we used an H_2_O_2_ solution, 808 nm laser, and UV laser irradiation to treat tumours in vitro and analysed the onset of immunogenic cell death (ICD). The immunofluorescence staining revealed bright green fluorescence (FL) on the membrane of the HeLa cells treated with Fe_3_O_4_@TiO_2_ + 808 nm, Fe_3_O_4_@TiO_2_ + UV, Fe_3_O_4_@TiO_2_ + H_2_O_2_, and Fe_3_O_4_@TiO_2_ + H_2_O_2_ + 808 nm + UV laser irradiation, indicating the induction of the ICD effect from the combination therapy (Figure 8a).

To examine the generation of ROS in HeLa cells following different treatments, we conducted DCFH-DA staining assays. Increased green fluorescence was observed in UV-irradiated cells, indicating ROS formation (Figure 8b). The H_2_O_2_-treated HeLa cells exhibited notably elevated fluorescence intensity compared to the control, attributed to H_2_O_2_ conversion into •OH radicals via Fe^2+^ catalysis in the Fenton reaction. Particularly noteworthy was the substantial increase in green fluorescence in the Fe_3_O_4_@TiO_2_ + H_2_O_2_ + 808 + UV-treated cells, indicating a remarkable rise in cellular ROS production. This outcome underscores the combined impact of CDT and PDT, which collectively amplified ROS generation significantly beyond what PDT alone could achieve.

## 4. Conclusions

In this study, we explored a smart Fe_3_O_4_@TiO_2_/DOX drug delivery platform for the PTT/PDT/chemotherapy/CDT combination treatment of cancer. The Fe_3_O_4_@TiO_2_/DOX microspheres demonstrated pH and NIR dual-responsive drug release behaviour in vitro. In addition, it has been proven that Fe_3_O_4_@TiO_2_ has superior biocompatibility. Fe^2+^ serves as a catalyst in the Fenton reaction, leading to the generation of •OH from H_2_O_2_ to facilitate CDT. Moreover, Fe_3_O_4_@TiO_2_ has been shown to catalyse the conversion of H_2_O_2_ into O_2_ within the cytoplasm, consequently enhancing the efficacy of PDT. In drug targeting systems, magnetically sensitive Fe_3_O_4_@TiO_2_/DOX containing medications can accurately accumulate at the tumour site with the help of an external magnetic field. Therefore, Fe_3_O_4_@TiO_2_/DOX has significant potential for the combination therapy of tumours.

An 808 nm laser possesses excellent tissue penetration, capable of penetrating several millimetres to centimetres into human tissues. UV light can penetrate through the epidermis into the dermis [42]. Both wavelengths can be applied to melanoma, taking advantage of their ability to reach the targeted tissues effectively. Additionally, Fe_3_O_4_@TiO_2_/DOX microspheres with a diameter of approximately 1 μm can be employed for interventional therapy in cancer.

## Data Availability

The data presented in this study are available on request from the corresponding author. The data are not publicly available due to privacy.

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
