# Peer review of "Fe3O4@TiO2 Microspheres: Harnessing O2 Release and ROS Generation for Combination CDT/PDT/PTT/Chemotherapy in Tumours"

_nanomaterials, 2024, doi:10.3390/nano14060498_

Round 1

Reviewer 1 Report

Comments and Suggestions for Authors

This article by Wang, Liang et al. describes the synthesis of Fe3O4@TiO2 microspheres with anticancer applications in mind. The article is overall very difficult to read. The figures and schemes are of low quality, sometimes impossible to read. The experimental procedures are poorly described, some protocols are simply not there, and the reader cannot, based on the provided information, hope to ever reproduce any of the experiments reported here. Many concepts seem unclear to the authors and the conclusions are not supported by the experimental data. I cannot encourage the publication of this article in Nanomaterials.

 List of specific comments (not exhaustive):

“Small molecule organic PSs, including sulfo-cyanine7 (Cy7), indocyanine green (ICG) and porphyrin, are employed as PSs for their excellent high ROS yields, while properties such as high toxicity and poor light absorption limit their bio-medical applications[6]. In contrast, inorganic materials such as TiO2 have shown long-term physical and chemical stability without bio-toxicity[7].”

- This statement is definitely subjective and the cited references are not convincing and do not support such a biased point of view.

«•O2-» should be O2-

 “The core-shell microspheres have a nearly flat surface”

- TEM images (Figure 1d) show a very rough surface, just similar to that of Fe3O4. SEM images also look similar between the two, except that 1e is blurred… Not convincing at all.

“diameter around 400-600 nm as shown in Figure 1d-g. These diameters were almost in agreement with those measured from SEM and TEM images”

- Actually TEM and SEM both seem to indicate that the average diameter is 800-1000 nm. Maybe showing more than one microsphere and evaluating the average size on a representative sample (measuring 250 spheres on TEM and SEM images) could help.

- It’s surprising that the EDS mapping does not show the core-shell nature of the constructs, especially when it comes to Ti mapping. Any thoughts?

- How do the authors explain such a high carbon content in TiO2-coated ferrite particles? Please provide the molar fraction of each element (this can be done easily based on EDS) and explain why there is C and N in the particles.

“Under 808 nm laser irradiation, the production of 1O2 by Fe3O4@TiO2 was assessed using DPBF (1,3-diphenylisobenzofuran). While the free DPBF exhibited no significant alteration in its UV-Vis absorbance spectra, Fe3O4@TiO2 displayed a rapid decrease in DPBF absorption upon UV laser exposure, indicating a swift generation of 1O2”

- So what is it? NIR, UV, visible light? So many contradictions in the same paragraph… 808 nm is definitely not UV. There is no clear experimental procedure given in the article. This is a big oversight.

“Figure 4f displayed the temperature of Fe3O4@TiO2 (1 mg mL-1, 1W cm2) increases by 51.1 â„ƒ after 5 min of exposure to NIR laser radiation. In comparison, Fe3O4 demonstrated a temperature change of 35.3 â„ƒ, whereas Fe3O4@TiO2 showed a temperature change of 29.2 â„ƒ under identical NIR irradiation conditions (Figure 4g). »

- Again, what is it? 51.1 °C or 29.2 °C for Fe3O4@TiO2? The text is contradictory and the figures are unclear.

“The DOX loading content (LC) of DOX in Fe3O4@TiO2 is 85.6%.”

- Loading content and loading efficiency are two very different parameters. This value is most likely loading efficiency (I doubt that DOX can account for 85% of the construct).

“Figure 6 depicts CLSM images of HeLa cells treated with FTTC-loaded microspheres for 8 hours, showcasing distinct blue fluorescence from DAPI staining for cell nuclei and green fluorescence from labelled FITC, with and without magnet field treatment. The Fe3O4@TiO2 were magnetically manipulated by the external magnetic field, showing an obvious green fluorescence signal overlapped with both the blue fluorescence signal. While the Fe3O4@TiO2 group without magnetic field loading had a weak fluorescence signal in the tumour site at 6h co-incubation. The results presented in Figure 6 indicate that the magnetic field enhanced the internalization of Fe3O4@TiO2, demonstrating its potential as an effective delivery system for targeted apoptosis induction in HeLa cells.”

- It is really difficult to make sense out of this section. How was FITC loaded within the microspheres? How were the particles externally “manipulated” within the cell culture? What “tumor site” are the authors talking about (how there be any tumor in 2D cell cultures)? How is apoptosis induction demonstrated here? Figure 6 does nothing to help and is not conclusive at all.

-  Figure 7 is too low definition to be legible. There is mention of control experiments. How do cells react to 30 min UV irradiation alone?

“When HeLa cells treated with Fe3O4@TiO2/DOX (1 mg mL-1) were exposed to the combined UV light+H2O2+808 nm group, their survival rate notably decreased to 7.8%. This rate was significantly lower compared to cells treated solely with UV light (32.32%), H2O(31.98%), or 808 nm laser (26.34%).”

- Are these latter values for cells treated under these conditions without Fe3O4@TiO2/DOX? This is unclear. The whole section is a challenge to understand, as most of the article’s content, actually.

- There is no discussion on the different wavelengths required for the different effects to be triggered, and their compatibility with the foreseen applications (in vivo).

- There is no discussion either on the size of the particles. How do the authors expect 1 µm large particles to behave in vivo? What kind of mobility can they hope for in biological environments such as the extra-cellular matrix, for instance?

- The authors claim that the particles have “superior biocompatibility” based on cytotoxicity assays, but without proper surface coating, what fate will such particles have in the blood stream (opsonization etc.)?

Comments on the Quality of English Language

The manuscript should definitely be edited by a native English speaker.

Reviewer 2 Report

Comments and Suggestions for Authors

This is an interesting paper that merits publication. It is clear and well written.

Only one correction:

Please, correct caption Fig1: f is missing

Reviewer 3 Report

Comments and Suggestions for Authors

The article by Zhao and co is very interesting, very clear, well written and certainly deserves publication. However, there are some things that need to be changed for its acceptance.

Introduction:

line 38-39: the authors mention three classes of compounds used as PS, but the field of PS is much broader.

From line 63 onwards: when the authors explain the purpose of their article they never mention the preparation of the microspheres with DOX, which therefore is not clear, at this level, why it was prepared.

Materials and methods:

in the paragraphs the symbols used for the various units of measurement must be reviewed which are often reported with the full name and not with the indication in the SI. For example, hours = h; grams = g; milligrams = mg; minutes = min; milliliters = mL

Results and discussion:

In general, a bit of discussion is missing when comparing the data obtained by the authors with the existing literature.

In particular:

line 160: it is said that the preparation of the microspheres takes place in three steps but in the text only the preparation of the Fe3O4 microspheres is explained, there is no mention of the step to obtain those containing titanium, much less those containing DOX.

Figure 1a is not cited in the text

Line 205 talks about the indirect method with DPBF of which the results are shown but this method is not reported in the materials and methods

Line 218 in the text figure 3h is mentioned but in reality, it is figure 3e

From line 233 to line 239 the term in vitro used is inappropriate

Line 295 talks about cellular uptake but even in this case the procedure is not reported in materials and methods

Line 304 “for targeted apoptosis induction in HeLa cells” this sentence has no specific confirmation from the data obtained by the authors

Line 321 is missing a control test of only the activity with H2O2 to verify that there is an actual increase

Line 331 the authors talk about synergistic treatment but to use the term synergy a specific test should be carried out

Figure 7b and 7c are difficult to read, it is not possible to understand exactly the tests that were carried out due to the low resolution of the images. In this sense, the text does not mention a test carried out with DOX alone to verify the improvement in system performance. Perhaps it is present in the figure but it is incomprehensible and in any case the result is not mentioned in the text of the article.

​

Round 2

Reviewer 1 Report

Comments and Suggestions for Authors

The authors have improved their manuscript and, even thogh not perfect, it is now better suited for publication.

Reviewer 3 Report

Comments and Suggestions for Authors

I believe that the authors have responded correctly to the various questions posed. The article has been properly revised and its quality has greatly improved